# Point Cloud Reconstruction Is Insufficient to Learn 3D Representations

## ABSTRACT

This paper revisits the development of generative self-supervised learning in 2D images and 3D point clouds in autonomous driving. In 2D images, the pretext task has evolved from low-level to high-level features. Inspired by this, through explore model analysis, we find that the gap in weight distribution between self-supervised learning and supervised learning is substantial when employing only low-level features as the pretext task in 3D point clouds. Low-level features represented by **PoI**nt **C**loud recons**T**ruction are ins**U**fficient to learn 3D **RE**presentations (dubbed **PICTURE**). To advance the development of pretext tasks, we propose a unified generative self-supervised framework. Firstly, high-level features represented by the Seal features are demonstrated to exhibit semantic consistency with downstream tasks. We utilize the Seal voxel features as an additional pretext task to enhance the understanding of semantic information during the pre-training. Next, we propose inter-class and intra-class discrimination-guided masking ($I^2$Mask) based on the attributes of the Seal voxel features, adaptively setting the masking ratio for each superclass. On Waymo and nuScenes datasets, we achieve 75.13% mAP and 72.69% mAPH for 3D object detection, 79.4% mIoU for 3D semantic segmentation, and 18.4% mIoU for occupancy prediction. Extensive experiments have demonstrated the effectiveness and necessity of high-level features. The project page is available at https://anonymous-picture.github.io/.

## CCS CONCEPTS

• **Computing methodologies → Unsupervised learning**; **Computer vision representations**.

## KEYWORDS

Self-supervised Learning, Autonomous Driving, Point Cloud Scene Understanding, Multimedia Foundation Models

## 1 INTRODUCTION

LiDAR has received widespread attention for its ability to simulate the depth and spatial distribution of the surrounding environment with high quality. In outdoor autonomous driving (AD), many works are based on LiDAR to achieve 3D perception, such as Point-Pillars [20], PV-RCNN [32], and SphereFormer [19]. Advanced 3D perception algorithms can significantly improve the safety of vehicles. However, a large amount of carefully annotated 3D data, such

*ACM MM, 2024, Melbourne, Australia*
© 2024 Copyright held by the owner/author(s). Publication rights licensed to ACM.
ACM ISBN 978-x-xxxx-xxxx-x/YY/MM
https://doi.org/10.1145/nnnnnnn.nnnnnnn

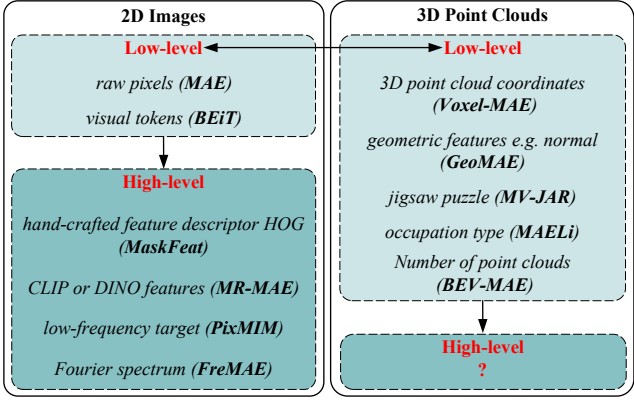

**Figure 1: Differences in the evolution process of pretext tasks between 2D images and 3D point clouds.**

as 3D boxes or point cloud semantic categories, is costly and time-consuming [26]. This leads to advanced models being trapped in fitting specific scenarios, and performance deteriorates significantly when migrating to other scenarios [17]. Learning the universal representation of 3D point clouds from a large amount of unlabeled data is a promising solution to alleviate the problems above.

Generative self-supervised learning (SSL), represented by masked autoencoders [13], has attracted attention in autonomous driving community. Occupancy-MAE [28] predicted whether each masked voxel was occupied. This promoted scene understanding by inferring the composition information of 3D scenes. The reconstruction target of GD-MAE [42] was to reconstruct the 3D coordinates of the point clouds in each masked voxel. GeoMAE [36] further introduced geometric features, e.g., normal, to enhance spatial understanding. MV-JAR [40] combined jigsaw and masked autoencoders to understand the spatial relationship between voxels. However, these self-supervised methods did not perform as expected on downstream tasks. Let's revisit the development process of representation learning in 2D images, as shown in Fig. 1. Low-level features are defined as unimodal, simple physical properties or spatial composition information, whereas the opposite is defined as high-level features. At the beginning, MAE [13] explicitly reconstructed missing raw pixels in low-level. Subsequently, MaskFeat [39] and MR-MAE [11] et al. reconstructed high-level features such as histogram of oriented gradient (HOG) and CLIP features. This has been proven to facilitate the interpretation of the image. However, current generative self-supervised learning methods for 3D point clouds still reconstruct explicit spatial information in low-level such as 3D coordinates. We argue that *the representation learning in 3D point clouds is equivalent to the primitive stage of 2D images.*

Downstream tasks, such as semantic segmentation and occupancy prediction, require encoders derived from self-supervised

learning to be proficient in extracting semantic information. This includes not only the multimodal features of single point cloud, such as images and text, but also the collective information from contextual point clouds. However, reconstructing spatial information focuses only on the 3D coordinates of the single point cloud, which is detrimental to downstream tasks that require semantic information. We believe that **P**o**I**nt **C**loud recons**T**ruction is ins**U**fficient to learn 3D **RE**presentations (dubbed **PICTURE**).

The question we ask is this: Why did representation learning in 3D point clouds not transition from explicit low-level features to implicit high-level features like in 2D images? Are implicit high-level features not applicable in 3D point clouds, or have appropriate methods not been found yet?

Recently, contrastive learning-based methods such as SLidR [31] and Seal [24] have explored collaborative learning on point cloud-image pairs obtained from superpixels [1] or vision foundation model SAM [16]. Through Seal, information is aggregated not only about the point cloud itself but also about the color of contextual points. From a qualitative perspective, we have demonstrated the semantic consistency between the Seal feature heatmap and the ground truth in downstream tasks through visualization. From a quantitative perspective, an exploratory model analysis based on weight distribution has been employed to demonstrate the promotion of using Seal features as pretext tasks on downstream tasks. Based on these observations, we propose a unified framework for generative self-supervised learning, **PICTURE**. We consider reconstructing Seal voxel features as an additional pretext task. Specifically, MinkUNet [7] pre-trained by Seal acts on raw point clouds. The point cloud features within the same voxel are aggregated to obtain Seal voxel features of the masked voxel. Reconstructing Seal voxel features implicitly benefits from image supervision and encourages the model to learn high-level vision concepts. More efficiently, we complete this process offline to accelerate the training.

The mask sampling strategy is essential to the generative SSL. Previous random masking treated all non-empty voxels equally. This neglected the varying difficulties caused by the discrimination of voxel features. Moreover, it dispersed the focus evenly across all regions instead of the categories of interest for downstream tasks. Inspired by Focal Loss [22], AttMask [15], etc., we develop a mask sampling strategy that is tightly coupled with the target Seal voxel features to adjust the difficulty. Specifically, we propose inter-class and intra-class discrimination-guided masking ($I^2$Mask). Firstly, eight superclasses that can represent autonomous driving scenarios are obtained unsupervised. On the one hand, for inter-class, we propose Fastest Class Sampling (FCS), which divides eight superclasses into three groups based on discriminative cues and sets base mask ratios. On the other hand, we define the intra-class consistency coefficient for each superclass and modulate the base mask ratio. As a result, the visualization of the mask ratio reveals that the reconstruction is more focused on regions like vehicles.

We conduct experiments on Waymo [34] and nuScenes [4] to verify promotion for downstream tasks. Compared to other self-supervised methods, our method achieves state-of-the-art results.

The main contributions of this paper are as follows:

- We propose a unified framework for generative self-supervised learning, PICTURE, in autonomous driving. Seal features with strong semantic information serve as additional reconstruction targets. 3D representation learning has been pushed forward following the developmental trajectory of 2D images.

- We propose inter-class and intra-class discrimination-guided masking. We have demonstrated that, compared to random masking, $I^2$Mask tightly coupled with target features achieves significant improvements.

- We achieve improvements of 30.4%, 26.1%, and 40.0% compared to advanced self-supervised methods in downstream tasks such as object detection, semantic segmentation, and occupancy prediction. We demonstrate the benefits through ablation studies.

## 2 RELATED WORK

### 2.1 Self-supervised Learning in AD

Learning universal features from large-scale point clouds in a self-supervised manner attracts attention in the autonomous driving community. Similar to 2D images, contrastive and generative self-supervised learning are two mainstream approaches. In contrastive self-supervised learning, ProposalContrast [45] applied random geometric transformations under two views to construct training samples. SLidR [31] and Seal [24] performed contrastive learning between corresponding images and point clouds through SLIC [1] or SAM [16]. CLIP$^2$ [47] followed the architecture of CLIP [30] and performed triple contrastive learning between images, point clouds, and text. TARL [29] required the encoder to extract equivalent features at different timesteps. In generative self-supervised learning, point cloud reconstruction is considered a crucial avenue to understanding point cloud scenes. 3D coordinates [40, 42] and other physical properties [14, 18, 36] were used as reconstruction targets. This method encourages the model to understand the details and distribution of 3D point clouds, which is the focus of this paper. Recently, drawing inspiration from language models and content generation, ViDAR [44] forced the model to predict the future from history and supervised the scene flow. UniPAD [43] rendered 2D RGB-D images using raw sparse point clouds.

### 2.2 Pretext Tasks for Generative SSL in AD

Pretext tasks are exploited by generative self-supervised methods to extract information from unlabeled datasets. The most common pretext task was to reconstruct the 3D coordinates of the point clouds in each masked voxel [14, 23, 40, 42]. Some work [3, 27, 28, 36] predicted whether each masked voxel was occupied. In addition, some other spatial information was used for pretext tasks, such as geometric features [36], jigsaw [40], occupation type [14, 18], and number of point clouds [14, 23]. However, the current pretext task in autonomous driving scenes is still focused on reconstructing explicit spatial information in low-level. Implicit semantic features in high-level have never been introduced as pretext tasks. We believe that explicit spatial information in low-level is insufficient.

### 2.3 Mask Sampling Strategy in Generative SSL

The mask sampling strategy determines where pretext tasks are applied. For 3D point clouds in autonomous driving, most of the work [3, 14, 36, 40] used random masking. MAELi [18] proposed that the masking ratio should decrease with the distance from ego, which not only reduced the difficulty of reconstructing far-range

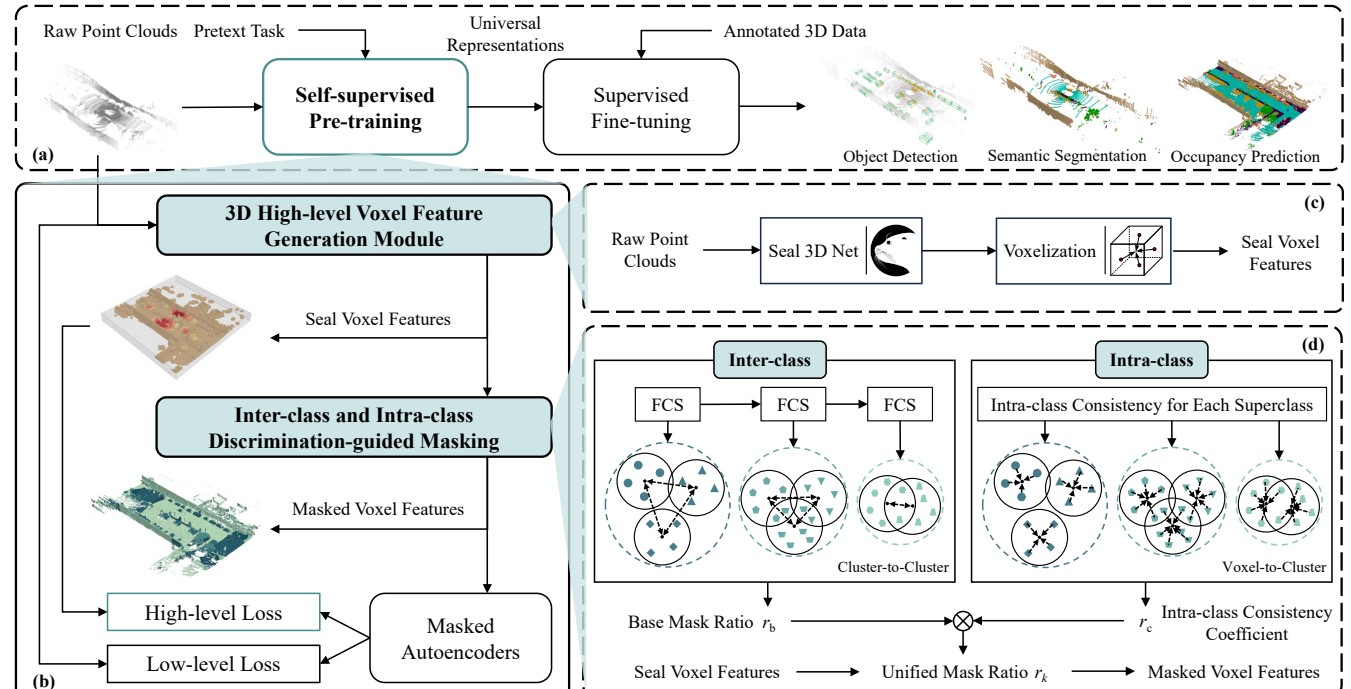

Figure 2: (a) The self-supervised learning process in 3D point clouds. (b) Architecture overview of unified generative self-supervised framework PICTURE. The raw 3D point clouds are fed into (c) the 3D high-level voxel feature generation module to obtain Seal voxel features. (d) Inter-class and intra-class discrimination-guided masking strategy is applied based on the attributes of Seal voxel features.

point clouds, but also ensured a consistent sparsity among both near-range and far-range points to enhance generalization ability of the model. BEV-MAE [23] applied a lower masking ratio in sparse areas to ensure a more stable reconstruction process. GD-MAE [42] implemented separate masking strategies at different granularities to maintain a consistent masking scope. However, mask sampling strategies based on feature attributes are less discussed in 3D generative SSL. In 2D images, AttMask [15] employed the attention map to establish the masking map. Areas that were highly attended were masked, making the reconstruction more efficient. SemMAE [21] and AMT [25] utilized iteration and throwing to make the attention map focus more on objects, respectively. Inspired by this, we develop a mask sampling strategy based on the attributes of the target Seal voxel features to adjust the difficulty and focus.

## 3 PROPOSED METHOD

In this section, we first introduce the unified self-supervised learning framework PICTURE in Fig. 2 and the process of exploiting Seal features in 3D high-level voxel feature generation module (Sec. 3.1). We then propose inter-class (Sec. 3.2) and intra-class (Sec. 3.3) discrimination-guided masking based on the feature attribute.

### 3.1 Architecture Overview of PICTURE

**Voxelization and Masking.** The point cloud can be defined as $\mathcal{P} = \left\{ c_\ell, f_\ell \middle| \ell = 1, \ldots, N_\mathrm{p} \right\}$, where $c_\ell$ and $f_\ell$ represent coordinates and point cloud features. We use the Voxel Feature Encoding (VFE) [41] for voxelization. The coordinates and features of all non-empty voxels are defined as $C_\mathrm{all} \in \mathbb{R}^{N_\mathrm{v} \times 3}$, $\mathcal{F}_\mathrm{all} \in \mathbb{R}^{N_\mathrm{v} \times C}$. The masking sampling strategy will be described in Sec. 3.2 and 3.3. The coordinates of the mask voxels are denoted as $C_\mathrm{m}$. The coordinates and features of unmasked voxels are denoted as $C_\mathrm{um}$, $\mathcal{F}_\mathrm{um}$.

**Sparse Encoder and Decoder.** To reduce the computational complexity of the attention, SST [9] and DSVT [37] specifically designed grouping mechanisms for sparse voxels. The coordinates $C_\mathrm{um}$ and feature $\mathcal{F}_\mathrm{um}$ of unmasked voxels are input into DSVT to extract features. The features of masked voxels are replaced by mask tokens and input into the decoder together with the output of the encoder. We also use DSVT as the decoder. In this way, the features of the masked voxels can be reconstructed.

**Reconstruction Target.** Reconstructing spatial features is not sufficient for downstream tasks that require semantic information. Seal [24] transferred image features from pre-trained vision foundation models to 3D networks, progressively aggregating contextual point clouds as it delves deeper into layers. This process can provide favorable semantic feature reconstruction signals for generative self-supervised learning. The Seal feature heatmaps [24] for example point cloud scene in nuScenes [4] are shown in Fig. 3(a). At the same time, the ground truth of the same scene in three downstream tasks are visualized in Fig. 3(b). It can be seen Seal features and the real scene have a high semantic consistency. The Seal feature is strong for road-related objects. Besides, the feature decreases

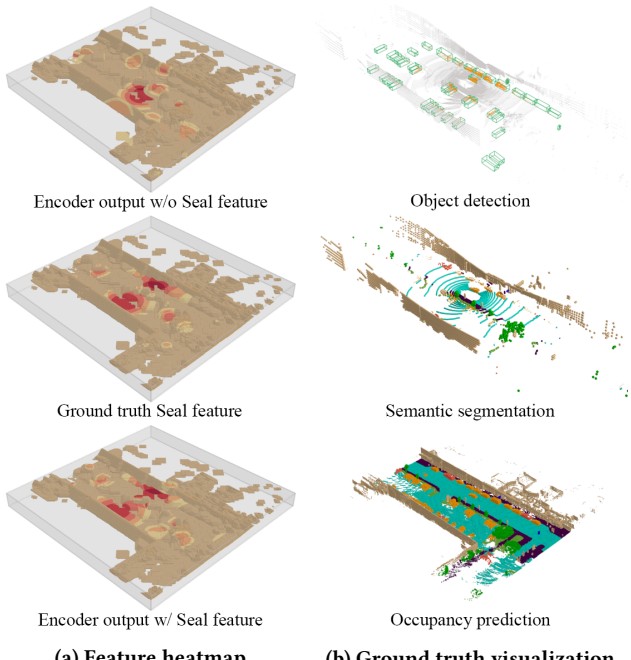

Encoder output w/o Seal feature

Object detection

Ground truth Seal feature

Semantic segmentation

Encoder output w/ Seal feature

Occupancy prediction

**(a) Feature heatmap**

**(b) Ground truth visualization**

Figure 3: (a) Heatmap for ground truth Seal features and encoder output pre-trained w/o and w/ Seal features. (b) Ground truth visualization for detection, segmentation, and occupancy prediction.

with distance from ego. Based on this observation, we consider reconstructing Seal voxel features as an additional pretext task.

Specifically, all raw point clouds $\mathcal{P}$ that have not been voxelized are fed into MinkUNet (Res16UNet34C) [7] pre-trained by Seal. Each point cloud aggregates spatial and feature information from other point clouds. We obtain the Seal point features $\mathcal{P}_s$. Subsequently, voxelization and average pooling are employed to aggregate sparse features that are located in the same voxel. Finally, we obtain the Seal voxel features $\mathcal{S}_{\text{all}} = \{s_j \mid j = 1, \ldots, N_v\}$ of all non-empty voxels. This process can be completed offline in advance to accelerate training. For masked voxels, the loss between reconstructed and target Seal voxel features is defined as:

$$L_{\text{seal}} = \text{SmoothL1}\left(\mathcal{S}_m^{\text{pred}}, \mathcal{S}_m^{\text{target}}\right). \tag{1}$$

Fig. 3(a) shows the heatmap extracted by the transformer encoder before and after introducing the pretext task. After introducing Seal voxel features, the features extracted by encoder have stronger semantic information, which is crucial for downstream tasks.

## 3.2 Inter-class Discrimination-guided Masking

We develop a mask sampling strategy based on the target Seal voxel features. For 22 categories in SemanticKITTI [2], 17 categories in nuScenes Lidarseg [10], and 22 categories in Waymo Semantic Segmentation [34], we first classify them into eight semantic classes: ground-related, structures, vehicle, two-wheeled vehicle, nature, human, object, and outlier. Semantic labels are agnostic in self-supervised learning. Therefore, we cluster Seal voxel features

---

**Algorithm 1** Fastest Class Sampling

**Input:** The set of all superclasses: $\mathcal{K}$
**Parameter:** Expected number of partition $n_1, n_2, n_3$
**Output:** The set of superclass partition $\mathcal{K}_1, \mathcal{K}_2, \mathcal{K}_3$
1: Let $\mathcal{K}_1, \mathcal{K}_2, \mathcal{K}_3 = \{\}, t = 1.$
2: **while** $t \leq 3$ **do**
3:    $\{k_1, k_2, \ldots, k_{n_t}\} \leftarrow$ select any $n_t$ superclass from $\mathcal{K}$.
4:    compute the distance between clustering centers of any two superclasses: $d(\boldsymbol{\mu}^{k_m}, \boldsymbol{\mu}^{k_n}) = \frac{1}{2}(1 - \boldsymbol{\mu}^{k_m}\boldsymbol{\mu}^{k_n})$.
5:    compute set of average inter-class distances $\mathcal{D}_{\text{inter}}$.
6:    $\mathcal{K}_t \leftarrow$ select superclass partition $\{k_1, k_2, \ldots, k_{n_t}\}$ with the fastest average inter-class distance $\max_{d \in \mathcal{D}} d$.
7:    $\mathcal{K} = \mathcal{K} \setminus \mathcal{K}_t$.
8:    $t = t + 1$.
9: **end while**
10: **return** $\mathcal{K}_1, \mathcal{K}_2, \mathcal{K}_3$

---

$\mathcal{S}_{\text{all}}$ using the k-means [6] and the number is set to 8. We obtain eight superclasses $k_i$. Although it is impossible to determine the correspondence between $k_i$ and semantic classes, this partition has considered the semantic attributes of the Seal voxel features.

We set the base masking ratio for each superclass based on the inter-class discrimination. First, we compute the cluster center $\boldsymbol{\mu}^{k_i}$, i.e., the mean of Seal voxel features:

$$\boldsymbol{\mu}^{k_i} = \frac{\sum_{j=1}^{N_{k_i}} \boldsymbol{s}_j^{k_i}}{N_{k_i}}, \tag{2}$$

where $N_{k_i}$ is the number of voxels of $k_i$. Next, we perform Algorithm 1 Fastest Class Sampling on clustering centers $\boldsymbol{\mu}^k$. We define the set $\mathcal{K} = \{k_i \mid i = 1, 2, \ldots, 8\}$. Then we select any $n_1$ superclass from $\mathcal{K}$ and calculate the distance between the cluster centers. The set $\mathcal{D}_{\text{inter}}$ composed of average inter-class distances is:

$$\mathcal{D}_{\text{inter}} = \left\{ \frac{1}{\binom{n_1}{2}} \sum_{m=1}^{n_1-1} \sum_{n=m+1}^{n_1} d(\boldsymbol{\mu}^{k_m}, \boldsymbol{\mu}^{k_n}) \,\middle|\, \{k_1, k_2, \ldots, k_{n_1}\} \in \binom{\mathcal{K}}{n_1} \right\}, \tag{3}$$

where $d(\boldsymbol{\mu}^{k_m}, \boldsymbol{\mu}^{k_n})$ represents the distance between the clustering centers of two superclasses, e.g., cosine similarity. Next, find the $n_1$ superclass partition with the fastest average inter-class distance, denoted as $\mathcal{K}_1$. Subsequently, the set difference between the original set $\mathcal{K}$ and $\mathcal{K}_1$ is used as the input for the next cycle. Select any $n_2$ superclass from the remaining set, repeat the above process, and finally obtain the $n_2$ superclass partition with the second fastest average inter-class distance, denoted as $\mathcal{K}_2$. The average inter-class distance of the remaining set is the smallest, denoted as $\mathcal{K}_3$.

The average inter-class distance reflects inter-class discrimination. Larger inter-class differences facilitate more straightforward distinction and reconstruction, and are typically associated with dynamic objects such as vehicles. We set a higher base mask ratio to increase the difficulty and the focus on the reconstruction. Based on this principle, we set the base mask ratio $r_b^1, r_b^2, r_b^3$ for each set.

## 3.3 Intra-class Discrimination-guided Masking

The intra-class discrimination reflects the difficulty of reconstructing masked voxel features from unmasked voxels. First, we define

Table 1: Comparisons of 3D object detection between PICTURE and other self-supervised learning methods on Waymo val set. $\dagger$ indicates the results are from the original paper. $*$ presents re-implemented by OpenPCDet. 'Epochs' and 'Fraction' denote the pre-training epochs and dataset fraction used for pre-training. The improvement compared to training from scratch is indicated with red superscripts.

| Method | Epochs | Fraction | L2 (AP/APH)↑ | | | |
| --- | --- | --- | --- | --- | --- | --- |
| | | | Overall | Vehicle | Pedestrian | Cyclist |
| SST$^\dagger$ [9] | - | - | 68.50 / 65.54 | 64.96 / 64.56 | 72.38 / 64.89 | 68.17 / 67.17 |
| Occupancy-MAE (SST)$^*$ [28] | 30 | 100% | 70.15$^{+1.65}$ / 67.16$^{+1.62}$ | 68.46 / 67.86 | 72.53 / 65.12 | 69.47 / 68.50 |
| MV-JAR (SST)$^*$ [40] | 30 | 100% | 70.38$^{+1.88}$ / 67.37$^{+1.83}$ | 68.59 / 68.05 | 72.77 / 65.23 | 69.78 / 68.82 |
| GD-MAE (SST)$^\dagger$ [42] | 30 | 100% | 70.62$^{+2.12}$ / 67.64$^{+2.10}$ | 68.72 / 68.29 | 72.84 / 65.47 | **70.30 / 69.16** |
| **PICTURE (SST)** | 30 | 20% | 69.86$^{+1.36}$ / 66.93$^{+1.39}$ | 68.14 / 67.53 | 72.28 / 64.83 | 69.15 / 68.43 |
| **PICTURE (SST)** | 30 | 100% | **71.02$^{+2.52}$ / 68.03$^{+2.49}$** | **69.37 / 68.84** | **73.44 / 66.15** | 70.26 / 69.09 |
| DSVT$^\dagger$ [37] | - | - | 73.20 / 71.00 | 70.90 / 70.50 | 75.20 / 69.80 | 73.60 / 72.70 |
| Occupancy-MAE (DSVT)$^*$ [28] | 30 | 100% | 73.86$^{+0.66}$ / 71.78$^{+0.78}$ | 71.53 / 71.21 | 76.02 / 70.69 | 74.02 / 73.44 |
| MV-JAR (DSVT)$^*$ [40] | 30 | 100% | 74.37$^{+1.17}$ / 72.01$^{+1.01}$ | 72.15 / 71.53 | 76.44 / 70.93 | 74.52 / 73.58 |
| GeoMAE (DSVT)$^*$ [36] | 30 | 100% | 74.46$^{+1.26}$ / 72.05$^{+1.05}$ | 72.13 / 71.62 | 76.61 / 71.02 | 74.64 / 73.52 |
| GD-MAE (DSVT)$^*$ [42] | 30 | 100% | 74.68$^{+1.48}$ / 72.22$^{+1.22}$ | 72.39 / 71.81 | 76.77 / 71.23 | 74.88 / 73.63 |
| **PICTURE (DSVT)** | 30 | 20% | 73.84$^{+0.64}$ / 71.80$^{+0.80}$ | 71.66 / 71.35 | 75.88 / 70.52 | 73.98 / 73.53 |
| **PICTURE (DSVT)** | 30 | 100% | **75.13$^{+1.93}$ / 72.69$^{+1.69}$** | **72.93 / 72.45** | **77.18 / 71.66** | **75.27 / 73.96** |

Table 2: Comparisons of 3D object detection between PICTURE and other self-supervised learning methods on nuScenes val set.

| Method | Epochs | Fraction | mAP↑ | NDS↑ | mATE↓ | mASE↓ | mAOE↓ | mAVE↓ | mAAE↓ |
| --- | --- | --- | --- | --- | --- | --- | --- | --- | --- |
| DSVT$^\dagger$ [37] | - | - | 66.4 | 71.1 | 27.0 | 24.8 | 27.2 | 22.6 | 18.9 |
| MV-JAR (DSVT)$^*$ [40] | 72 | 100% | 67.6$^{+1.2}$ | 72.2$^{+1.1}$ | 26.5 | 24.4 | 26.8 | 22.1 | 18.3 |
| GD-MAE (DSVT)$^*$ [42] | 72 | 100% | 67.7$^{+1.3}$ | 72.2$^{+1.1}$ | 26.4 | 24.4 | 26.9 | 21.9 | 18.2 |
| **PICTURE (DSVT)** | 72 | 100% | **68.1$^{+1.7}$** | **72.6$^{+1.5}$** | **25.8** | **24.2** | **26.5** | **21.6** | **17.7** |

the set of the average intra-class distance $\mathcal{D}_{\text{intra}}$:

$$\mathcal{D}_{\text{intra}} = \left\{ \frac{\sum_{j=1}^{N_{k_i}} \mathbb{1}\{d(\boldsymbol{\mu}^{k_i}, \boldsymbol{s}_j^{k_i}) - \lambda > 0\} d^2(\boldsymbol{\mu}^{k_i}, \boldsymbol{s}_j^{k_i})}{\sum_{j=1}^{N_{k_i}} \mathbb{1}\{d(\boldsymbol{\mu}^{k_i}, \boldsymbol{s}_j^{k_i}) - \lambda > 0\}} \,\middle|\, i \in \{1, 2, \ldots, 8\} \right\}, \quad (4)$$

where $d(\boldsymbol{\mu}^{k_i}, \boldsymbol{s}_j^{k_i})$ represents the distance between the Seal features of a voxel and its clustering center. The $\mathbb{1}\{\}$ stands for indicator function. The denominator of Eq. (4) represents the number of voxels whose distance from the clustering center exceeds threshold $\lambda$. The meaning of $\mathcal{D}_{\text{intra}}$ is the average distance that is too far from the cluster center. This can reflect the consistency between Seal voxel features in a superclass.

We define intra-class consistency coefficient $r_{c_i}$ to reflect the intra-class discrimination:

$$r_{c_i} = 1 - \frac{\mathcal{D}_{\text{intra}}^i}{\max_{d \in \mathcal{D}_{\text{intra}}} d}. \quad (5)$$

The intra-class consistency coefficient $r_{c_i}$ for a superclass has the opposite trend to the average intra-class distance $\mathcal{D}_{\text{intra}}^i$. It is easier to reconstruct masked voxel features from unmasked voxels in regions with high intra-class consistency coefficient $r_{c_i}$. Therefore, we use $r_{c_i}$ to modulate the base masking ratio to obtain the unified masking ratio $r_{k_i}$ for each superclass:

$$r_{k_i} = (r_{\text{b}}^1 \cdot \mathbb{1}\{k_i \in \mathcal{K}_1\} + r_{\text{b}}^2 \cdot \mathbb{1}\{k_i \in \mathcal{K}_2\} + r_{\text{b}}^3 \cdot \mathbb{1}\{k_i \in \mathcal{K}_3\}) \cdot r_{c_i}. \quad (6)$$

Each superclass calculates the unified mask ratio uniquely based on its inter-class and intra-class discrimination.

## 4 EXPERIMENTS

### 4.1 Experimental Settings

**Dataset.** We demonstrate the performance of PICTURE in 3D object detection on Waymo Open Dataset [34], 3D semantic segmentation on nuScenes [4], and occupancy prediction on OpenOccupancy [38]. To mitigate the risk of data leakage, we perform pre-training and fine-tuning on the training set, and report the results on the val and test set. For semantic segmentation and occupancy prediction, we only report the Intersection over Union (IoU) for a subset of key objects.

**Model.** We utilize the popular frameworks OpenPCDet [35] and MMDetection3D [8]. We fine-tune the transformer-based encoders SST, DSVT, and Cylinder3D-SST. By default, we employ 4 DSVT (with 8 attention layers) as encoders and 2 DSVT (with 4 attention layers) as decoders. The feature dimension is set to 192. Regarding the number of encoders and decoder types, please refer to the

**Table 3: Comparisons of 3D semantic segmentation between PICTURE and other self-supervised methods on nuScenes val set.**

| Method | mIoU↑ | bicycle | bus | car | motorcycle | pedestrian | trailer | truck |
|---|---|---|---|---|---|---|---|---|
| Cylinder3D-SST[†] [9] | 76.5 | 40.0 | 91.8 | 94.2 | 78.1 | 80.1 | 62.5 | 84.7 |
| GeoMAE (Cylinder3D-SST)[†] [36] | 78.6[+2.1] | 42.6 | 93.5 | 95.8 | 79.8 | 83.5 | 65.6 | 87.3 |
| ALSO (Cylinder3D-SST)* [3] | 78.8[+2.3] | 42.4 | 93.8 | 95.5 | 80.2 | 83.7 | 65.6 | 86.8 |
| **PICTURE (Cylinder3D-SST)** | **79.4**[+2.9] | **43.2** | **94.5** | **96.3** | **80.6** | **84.1** | **65.7** | **87.5** |

**Table 4: Comparisons of occupancy prediction between PICTURE and self-supervised methods on OpenOccupancy val set.**

| Method | mIoU↑ | bicycle | bus | car | motorcycle | pedestrian | trailer | truck |
|---|---|---|---|---|---|---|---|---|
| DSVT* [37] | 16.3 | 6.4 | 13.8 | 18.5 | 5.6 | 10.1 | 13.8 | 14.2 |
| Occupancy-MAE (DSVT)* [28] | 17.2[+0.9] | 6.7 | 15.1 | 19.3 | 6.6 | 11.3 | 14.5 | 15.3 |
| GD-MAE (DSVT)* [42] | 17.8[+1.5] | 7.2 | 15.1 | 20.1 | 6.8 | 11.9 | 15.3 | 15.5 |
| **PICTURE (DSVT)** | **18.4**[+2.1] | **8.1** | **15.6** | **20.7** | **7.4** | **12.3** | **15.7** | **15.8** |

**Table 5: Comparisons of 3D object detection on Waymo test.**

| Method | L2 (AP/APH)↑ | |
|---|---|---|
| | **Overall** | **Vehicle** |
| CenterPoint [46] | 73.38 / 71.93 | 73.42 / 72.99 |
| SST [9] | 74.41 / 72.81 | 73.08 / 72.74 |
| DSVT [37] | 74.76 / 73.07 | 75.11 / 74.10 |
| PV-RCNN++ [33] | 75.00 / 73.52 | 76.31 / 75.92 |
| GD-MAE (SST) [42] | 76.47 / 73.37 | 75.83 / 75.46 |
| **PICTURE (DSVT)** | **77.52 / 76.10** | **77.70 / 77.30** |

supplementary material for more ablation experiments. For Waymo, we set the voxel size to [0.32, 0.32, 0.1875]. For nuScenes (semantic segmentation and occupancy prediction), we set it to [0.2, 0.2, 0.2].

***Training Details.*** In inter-class discrimination-guided masking, we set the expected number of superclass partition $(n_1, n_2, n_3)$ as (3, 3, 2) and the base mask ratio $(r_b^1, r_b^2, r_b^3)$ as (0.9, 0.45, 0), meaning that some superclasses are not masked. We set the distance threshold $\lambda$ in intra-class discrimination-guided masking to 0.6. For detailed ablation experiments on hyperparameters, please refer to the supplementary material. During pre-training, we employ AdamW and train for 30 epochs on Waymo and 72 epochs on nuScenes. For fine-tuning, we train for 12 epochs on Waymo and 24 epochs on nuScenes. The loss weights for reconstructing low-level and high-level features are set to 1.0 and 3.0, respectively. All experiments are conducted on 8 NVIDIA A100-SXM4-40GB GPUs.

## 4.2 Comparison with State-of-the-art Methods

We first compare the performance of our method with other self-supervised learning methods in 3D object detection. Tab. 1 and Tab. 5 present the performance on the Waymo val set and test set for the leaderboard, respectively. We employ two 3D transformer blocks, SST and DSVT, as encoders. The compared low-level features include 3D point cloud coordinates (Occupancy-MAE [28], GD-MAE [42]), geometric features (GeoMAE [36]), and the jigsaw puzzle (MV-JAR [40]). Our proposed PICTURE outperforms training from scratch, leading to improvements of 1.93% and 1.69% in

L2 mAP on the Waymo val set. Meanwhile, compared to the best self-supervised learning method, GD-MAE, our proposed PICTURE achieves improvements of 0.45% and 0.47%. This is attributed to the semantic consistency of high-level features and the feature attribute-related mask sampling strategy I[2]Mask. Furthermore, it can be observed that as the pre-training data increases, the performance significantly improves from 73.84% to 75.13%, opening up possibilities for leveraging large-scale unlabeled point clouds.

For 3D object detection, in addition to the Waymo dataset [34], we have also reported results on the val set of the nuScenes [4] in Tab. 2. Despite utilizing the robust DSVT as encoder, our proposed method, PICTURE, still manages to achieve improvements of 1.7% in mAP and 1.5% in NDS, respectively. Compared to using only low-level features such as 3D point cloud coordinates and jigsaw, employing high-level features for reconstruction targets results in improvements of 0.4% and 0.5% in mAP, respectively. This indicates that our proposed 3D self-supervised reconstruction target exhibits strong generalization across various autonomous driving datasets.

Tab. 3 provides a comparison of 3D semantic segmentation on the nuScenes val set. We consider low-level features such as geometric features and the occupation type specifically designed for segmentation (ALSO [3]). Our proposed PICTURE outperforms them by 0.8% and 0.6%, respectively, and exhibits greater advantages in categories like bicycle, car, and motorcycle. This effectively demonstrates the beneficial impact of high-level features on segmentation tasks that rely on semantic information.

Tab. 4 provides a comparison with other self-supervised methods in occupancy prediction. Since the Seal feature is aligned with both images and text, reconstructing the Seal voxel features can enhance the performance of occupancy prediction. Our proposed PICTURE surpasses GD-MAE by 0.6%. Furthermore, our proposed I[2]Mask places more attention on road-related objects, and improvement in occupancy prediction accuracy is crucial for driving safety.

## 4.3 Ablation Studies

Tab. 6 presents the ablation studies on the reconstruction target and mask sampling strategy. Firstly, with random masking, supplementing with Seal voxel features as a pretext task results in an

**Table 6: Ablation study of pre-training, reconstruction target, and mask sampling strategy on the Waymo val set.**

| Pre-train | Reconstruction target | $I^2$Mask | L2 mAP | L2 mAPH |
|-----------|----------------------|-----------|--------|---------|
| None | - | - | 73.20 | 71.00 |
| PICTURE | Coord. | × | 74.05 | 72.11 |
| | Coord. + Geo. | × | 74.26 | 72.25 |
| | Coord. + Seal | × | 74.74 | 72.56 |
| | Coord. + Seal | only inter-class | 74.82 | 72.60 |
| | Coord. + Seal | only intra-class | 74.85 | 72.58 |
| | Coord. + Seal | ✓ | **75.13** | **72.69** |

**Table 7: Ablation study of different high-level features for pretext tasks in 3D object detection on Waymo val set.**

| Pre-train | Reconstruction target | L2 mAP | L2 mAPH |
|-----------|----------------------|--------|---------|
| None | - | 73.20 | 71.00 |
| PICTURE | Coord. | 74.05 | 72.11 |
| | Coord. + SLidR [31] | 74.42 | 72.20 |
| | Coord. + CLIP$^2$ [47] | 74.70 | 72.42 |
| | Coord. + CLIP2Scene [5] | 74.62 | 72.26 |
| | Coord. + Seal [24] | **75.13** | **72.69** |

improvement of 0.69% and 0.48% in L2 mAP compared to various low-level features, such as 3D coordinates and geometric features. This indicates that the Seal feature exhibits a significant positive impact on downstream tasks owing to its high semantic consistency. Secondly, the performance further improves when replacing the random mask with $I^2$Mask. The base mask ratio $r_b$ and intra-class consistency coefficient $r_c$ obtained from inter-class and intra-class discrimination-guided masking further contribute to an additional improvement of 0.08% and 0.11%, respectively. When both distance metrics are jointly considered, distinct mask ratios are assigned to each superclass, resulting in an ultimate improvement of 0.39%.

In addition to Seal features, we also explore alternative high-level features that could be used for pretext tasks in Tab. 7. It can be observed that compared to low-level features, all high-level features can yield a minimum improvement of 0.37%. This indicates that using low-level features as the pretext task in 3D point cloud self-supervised representation learning is insufficient. Various high-level features can yield benefits, not just Seal features. On the other hand, although all point cloud encoders are aligned with the semantic information of images or texts through contrastive learning, Seal features can achieve the best alignment results by locating image-point cloud pairs via SAM. Seal features yield the best performance, with improvements ranging from 0.43% to 0.71% compared to other high-level features. We condense these observations and analyses into a concise implementation.

## 4.4 Data Efficiency

Comparison with training from scratch on different data scales for fine-tuning is shown in Tab. 8. On the one hand, at all data scales, the encoder pre-trained with PICTURE outperforms no pre-training.

**Table 8: Comparison with no pre-training on different data scale in 3D object detection using L2 mAPH on Waymo val.**

| DSVT w/ PICTURE | 10% | 20% | 50% | 100% |
|-----------------|-----|-----|-----|------|
| | 52.71 | 57.46 | 67.32 | 71.00 |
| ✓ | 56.62$^{+3.91}$ | 61.87$^{+4.41}$ | 70.56$^{+3.24}$ | 72.69$^{+1.69}$ |

Specifically, when there is less fine-tuning data, the advantage of pre-training becomes more pronounced. For instance, when fine-tuning with 20% of the data, there is a 4.41% increase in L2 mAPH. On the other hand, with only 50% fine-tuning data required, the performance of DSVT w/PICURE is close to training from scratch at 100% fine-tuning data, which is beneficial for autonomous driving communities that severely lack annotated data.

## 4.5 Study of Time Cost

The time cost of our proposed method, PICTURE, compared to other self-supervised learning methods, is presented in Tab. 9. Despite the superior performance of Seal features compared to low-level features, such as geometric features [36], it requires approximately 5× time cost (215h vs. 48h). This represents a trade-off between performance and time cost. To accelerate the pre-training, we adopt an offline approach to pre-extract Seal features for the entire dataset and save them locally. Thus, Seal features can be reused throughout pre-training. Following offline processing, the time cost for our PICTURE is approximately 55 hours, which is comparable to the time required for low-level features (55h vs. 61h, 48h).

## 4.6 Visualization of Mask Sampling Strategy

Fig. 4 shows the semantic and mask ratio distribution for a certain scene. Compared to randomly masking all non-empty voxels, our proposed $I^2$Mask assigns mask ratios with inverse trends for different regions based on their reconstruction difficulty. Moreover, it can be seen areas such as vehicles and pedestrians, that are highly focused on in downstream tasks, have a high masking ratio, which forces complex reconstruction. On the contrary, the masking ratio is lower for areas such as roads and constructions, which reduces the attention during the reconstruction.

## 5 EXPLORATORY MODEL ANALYSIS

Pretext tasks play a pivotal role in self-supervised learning. Appropriately designed pretext tasks guide the network towards the generalized features from the feature space. Further, the feature extraction may be quantitatively represented as weights. Therefore, we explore the differences between pretext tasks from the perspective of weight distribution. We select the weights of the value nodes in all attention layers of the encoder as our research subject, including the weights from multi-heads.

*What is the promotion of high-level features in 2D images?* We employ the reconstruction of raw pixels as a low-level pretext task and the reconstruction of CLIP feature as a high-level pretext task. To simplify the analysis, we assume the weights follow a Gaussian distribution [12] and compute the mean $\mu_l^{2d}, \mu_h^{2d}$ and variance $\sigma_l^{2\,2d}, \sigma_h^{2\,2d}$ using maximum likelihood estimation. We assume

**Table 9: Time cost of self-supervised learning methods during pre-training on 8 A100-SXM4-40GB GPUs on Waymo.**

| Method | Epochs | Fraction | Time |
|---|---|---|---|
| MV-JAR (DSVT) [40] | 30 | 100% | 61h |
| GeoMAE (DSVT) [36] | 30 | 100% | 48h |
| **PICTURE (online) (DSVT)** | 30 | 100% | 215h |
| **PICTURE (offline) (DSVT)** | 30 | 100% | 55h |

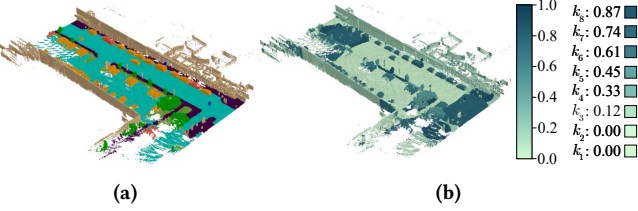

(a)                                     (b)

**Figure 4: (a) Ground truth visualization of occupancy prediction. (b) The mask ratio distribution of a certain scene derived by I²Mask.**

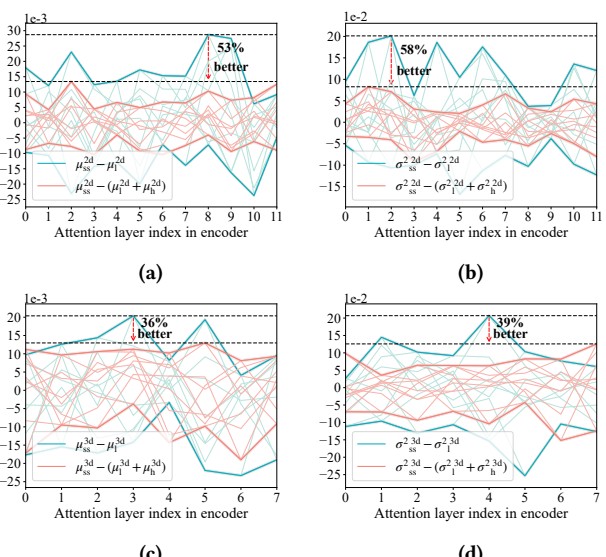

(a)                          (b)

(c)                          (d)

**Figure 5: The disparity in the weight distribution of each attention layer in the encoder. (a) and (b) represent 2D images, while (c) and (d) correspond to 3D point clouds.**

the weight distribution derived from supervised learning $\mu_s^{2d}, \sigma_s^{2\,2d}$ serves as the ideal weight distribution for self-supervised learning $\mu_{ss}^{2d}, \sigma_{ss}^{2\,2d}$. Fig. 5(a) and (b) visualize the disparity in the weight distribution of each attention layer, which are the mean and variance. We use differences to measure disparity without loss of generality. It can be observed that when only reconstructing low-level features, the amplitude of the differences is erratic, indicating that low-level features in 2D images are insufficient to guide the network in learning a weight distribution that is beneficial for downstream tasks. When high-level features are supplemented, the weight distribution approaches that of supervised learning.

*Are low-level features in 3D point clouds sufficient as pretext tasks?* The same analysis is applied to 3D point clouds. We use DSVT [37] as the encoder. We employ the reconstruction of 3D point cloud coordinates as a low-level pretext task, and the reconstruction of Seal voxel features as a high-level pretext task. The weight distribution derived from the downstream 3D object detection task $\mu_s^{3d}, \sigma_s^{2\,3d}$ serves as the ideal weight distribution for self-supervised learning $\mu_{ss}^{3d}, \sigma_{ss}^{2\,3d}$. Fig. 5(c) and (d) visualize the disparity from the ideal weight distribution when using only low-level features and when supplementing with high-level features. It can be observed low-level features are insufficient for learning 3D representations, as there are significant disparities in mean and variance. After supplementing with Seal voxel features, the parameters advance in a direction beneficial to downstream tasks. This quantitatively indicates the necessity and feasibility of introducing high-level features in self-supervised learning for 3D point clouds.

## 6 LIMITATION

**Firstly**, compared to existing methods, the improvement is not sufficiently significant. On the one hand, the capability of the high-level feature extractor, Seal, is restricted. Due to the lack of large-scale point cloud-image-text datasets, current high-level feature

extractors cannot leverage the powerful understanding abilities of large language models. On the other hand, open-source point cloud datasets in autonomous driving are significantly smaller compared to internet-scale datasets, which limits the full potential of self-supervised learning. **Secondly**, the framework may perform poorly in areas far from ego, where the point clouds are sparse. The masked voxels fail to reconstruct at excessively far distances due to the lack of nearby unmasked voxels. In our subsequent work, our proposed I²Mask can consider not only the difficulty of the sample but also the distance from the ego. **Finally**, the time cost of extracting Seal features is significantly higher compared to low-level features such as geometric features. We pre-extracte Seal features for all point cloud scenes in the entire dataset. Loading these features offline during pre-training allows the time cost of PICTURE to be similar to other self-supervised methods. However, the process of pre-extracting Seal features incurs a significant time cost.

## 7 CONCLUSION

Generative self-supervised learning in 3D point clouds is trapped in low-level features. We present the necessity of introducing high-level features from four perspectives: field development, consistency visualization with downstream tasks, quantitative experiments, and exploratory model analysis from the perspective of weight distribution. Reconstructing Seal voxel features during pre-training brings benefits to downstream tasks that require semantic information. Furthermore, in contrast to random masking, we propose inter-class and intra-class discrimination-guided masking (I²Mask) to adaptively set the masking ratio for each superclass, which explores the potential of the Seal voxel feature. Extensive experiments confirm our contributions to advancing generative self-supervised learning in 3D point clouds.

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
