# OpenReview forum: "Point Cloud Reconstruction Is Insufficient to Learn 3D Representations"
_acmmm.org/ACMMM/2024/Conference — MM2024 Poster_

### Official Review · Reviewer_FAZy · 2024-05-06

**Rating:** 4
**Confidence:** 4

**Summary:**

The paper titled "Point Cloud Reconstruction Is Insufficient to Learn 3D Representations' proposes a unified generative self-supervised framework for masked point modeling, which reconstructs both the raw coordinates and high-level features (extracted by Seal). Experiments on autonomous driving datasets showcase the effectiveness of designed framework and discrimination-guided masking strategy.

**Strengths:**

The paper is well-written, with exquisite figures and tables.

Both qualitative and quantitative experiments are comprehensive, particularly the creative visual analysis in Figure 5.

The article candidly analyzes the limitations of the proposed method, such as time consumption and accuracy improvement.

**Limitations:**

Main concerns:

1, The improvement is marginal, which is also pointed out by the authors in the Limitation section.

2, The discrimination-guided masking strategy in the paper is non-trainable, and its hyperparameters need to be manually specified. Moreover, Tables D.2, D.3, and D.4 in the Appendix indicate that different hyperparameters have a significant impact on the results.

Detailed explanations for the observations are welcomed:

3, When using only intra-class or inter-class distance metrics, the improvement is minimal in Table 6. However, when using both types of distance metrics, the improvement becomes significantly higher.

4, In Table 7, only when using seal features as the high-level feature can the results surpass the existing SOTA method (i.e., GD-MAE). The introduction of other high-level features does not achieve this effect.

5, In Table 8, when fine-tuning with 10% of the data, the incremental gain from pre-training is not as significant as when fine-tuning with 20% of the data. This seems somewhat counterintuitive.

**Suitability:**

2

---

### Official Review · Reviewer_GnT8 · 2024-05-25

**Rating:** 3
**Confidence:** 4

**Summary:**

This paper focus on pretraining a 3D network specifically for 3D point clouds in autonomous driving. It reconstructs the semantic features using the Seal pretrained network and integrates an existing spatial reconstruction loss into MAE-based pretraining method. This method enhances the MAE-based pretraining technique by facilitating the extraction of higher-level semantic features, addressing the limitations inherent in current generative self-supervised learning approaches for 3D point clouds. Traditional methods mainly focus on reconstructing explicit low-level spatial information, such as 3D coordinates, which can be limiting. This proposed enhancement seeks to overcome these constraints by improving the network’s ability to extract features at a higher semantic level. Meanwhile, it also introduces a feature-guided mask sampling strategy, inspired by recent works, such as Focal Loss, AttMask.

**Strengths:**

The strengths of this paper are threefold. First, the authors have made a significant stride in learning higher-level semantic features, which effectively address the limitations of spatial-based MAE that primarily learns low-level features. Additionally, the introduction of the farthest categories sampling is a noteworthy contribution. It overcomes the inefficacies of random masking strategies, which often retain unuseful points, making it difficult to reconstruct the raw point cloud accurately. Lastly, the numerical results demonstrate a significant improvement, underscoring the efficacy of the proposed methods.

**Limitations:**

The limitations of this study can be outlined in four main points. First, the high-level features reconstruction method presented is incremental and heavily relies on the existing pre-trained method, Seal. Although utilizing additional information from Seal can enhance performance, a novel method that does not depend on pre-existing pretrained features would be more innovative and is highly anticipated. Second, the paper lacks a direct comparison with Seal, despite utilizing its features. Providing a visual comparison of the features pre-trained with Seal would significantly enhance the credibility and depth of the analysis. Third, the ablation study is incomplete as it does not present results from experiments that exclusively use feature reconstruction, omitting those that also involve coordinates reconstruction. Finally, the masking strategy also depends on supplementary information derived from the Seal pretrained network, which may limit the generalizability of the approach.

**Suitability:**

2

---

### Official Review · Reviewer_9udE · 2024-05-25

**Rating:** 4
**Confidence:** 3

**Summary:**

To overcome the gap in weight distribution betweeen self-supervised learning and supervised learning when only low feature employed, the authors propose a unified generative self-supervised framework. The method proposed
by the authors mainly innovates in two ways:

1.Utilizing of the high-level features: high-level features are demonstrated to exhibit semantic consistency
with downstream tasks. The authors utilize the Seal voxel features as an additional pretext task to enhance the
understanding of semantic information during pre-training.

2. inter-class and intra-class discrimination-guided masking: based on the attributes of the Seal voxel
features, the inter-class and intra-class discrimination-guided masking adaptively setting the masking ratio
for each superclass.

**Strengths:**

1.Unlike previous methods that cannot Learning the universal representation of 3D point
clouds from a large amount of unlabeled data. PICTURE propose a unified framework for
generative self-supervised learning with seal features served as additional
reconstruction targets.

2.Compared to random masking, inter-class and intra-class discrimination-guided masking
proposed by the authors tightly coupled with target features and achieves significant
improvements.

3.The model achieves significant improvements compared to advance self-supervised
methods in downstream tasks.

**Limitations:**

1. The performance is a little weaker than the previous method in comparison of Cyclist.

2. in the part of the methods, the details for the theory explanation are limited.

**Suitability:**

2

---

### Official Review · Reviewer_kBrw · 2024-05-27

**Rating:** 3
**Confidence:** 3

**Summary:**

This paper discusses the limitations of generative self-supervised learning in the field of 3D point clouds, particularly in autonomous driving. It highlights the challenges associated with using "low-level features" for 3D point cloud reconstruction and proposes a new approach using "high-level features" in addition to "low-level features" to improve the learning of 3D representations.

**Strengths:**

1. This paper improves the basic reconstruction-based method by introducing a "high-level" loss. The proposed method uses the SEAL feature to gain more information during the pre-training process. I think such idea makes sense to me. Simply using the reconstruction method is not enough for point cloud pre-training.

2. The usefulness of the proposed method has been validated on a number of downstream tasks. There is some performance gain shown in Table 1 - Table 5, compared with GD-MAE. Also, I can see the effectiveness of inter-class / intra-class guided masking from the ablation experiments.

**Limitations:**

### Some flaws related with the paper
> current generative self-supervised learning methods for 3D point clouds still reconstruct explicit spatial information in low-level such as 3D coordinates

Actually, there are some papers about 3D point cloud representation learning (though not used for automatic driving scenario) that uses the "high-level" features.
[1] Xue, Le, et al. "Ulip: Learning a unified representation of language, images, and point clouds for 3d understanding."
[2] Yao, Yuan, et al. "3d point cloud pre-training with knowledge distillation from 2d images."

> Low-level features are defined as unimodal, simple physical properties or spatial composition information, whereas the opposite is defined as high-level features

I think the high-level features are not well-defined here. The opposite of unimodal features is clear to me (multi-modal features). However,  what is the opposite of simple physical properties / spatial composition seems vague. You should clearly define the high-level features you are referring to.

> We argue that the representation learning in 3D point clouds is equivalent to the primitive stage of 2D images

This statement should be validated. Are there experiments / some arguments pointing to this?

### Some general questions

1. Why do you think the inter-class and intra-class features are high-level features? What is their substantial difference from the low-level physical features?

2. If I understand your method correctly, you don't use class labels during pre-training. Instead you use class labels generated by unsupervised learning. Since you don't use class labels during pre-training, why do you think pre-training with "classes" works?

3. Do you think the features of Seal is good enough? From the heat map it seems that the Seal feature still has some disagreement with the ground-truth labels.

### Some minor errors:
> Why did representation learning in 3D point clouds not transition from explicit low-level features to implicit high-level features like in 2D images?

I don't understand this sentence.

> Firstly, eight superclasses that can represent autonomous driving scenarios are obtained unsupervised.

Where is secondly, thirdly etc.?

**Suitability:**

3

---

### Meta-Review · Area_Chair_3e41 · 2024-06-30

**Recommendation:** Accept (Poster)
**Confidence:** 4

**Metareview:**

All reviewers lean to acceptance of this submission. Authors please include revisions to comments from reviewers, e.g., over-claim, comprehensive literature etc